# Knowledge Gap or Prepared Force? Exploring United Arab Emirates Pharmacy Students and Pharmacists’ Monkeypox Readiness

**DOI:** 10.3390/healthcare12222295

**Published:** 2024-11-16

**Authors:** Razan I. Nassar, Alhareth Ahmad, Iman A. Basheti, Amin M. Omar, Hiba Jawdat Barqawi, Karem H. Alzoubi, Moyad Shahwan, AlMuzaffar M. Al Moukdad, Mays Alrim Al Moukdad, Eman Abu-Gharbieh

**Affiliations:** 1Department of Clinical Pharmacy and Therapeutics, Faculty of Pharmacy, Applied Science Private University, Amman 11931, Jordan; r_nassar@asu.edu.jo; 2Pharmacological and Diagnostic Research Centre, Faculty of Pharmacy, Al-Ahliyya Amman University, Amman 19328, Jordan; 3Pharmaceutical Sciences Department, Faculty of Pharmacy, Jadara University, Irbid 21110, Jordan; 4Department of Pharmaceutical Science and Pharmaceutics, Faculty of Pharmacy, Applied Science Private University, Amman 11931, Jordan; 5Research Institute of Medical and Health Sciences, University of Sharjah, Sharjah P.O. Box 27272, United Arab Emirates; 6Department of Clinical Sciences, College of Medicine, University of Sharjah, Sharjah P.O. Box 27272, United Arab Emirates; 7Department of Pharmacy Practice and Pharmacotherapeutics, College of Pharmacy, University of Sharjah, Sharjah P.O. Box 27272, United Arab Emirates; 8Department of Clinical Sciences, College of Pharmacy and Health Sciences, Ajman University, Ajman P.O. Box 346, United Arab Emirates; 9Centre of Medical and Bio-Allied Health Sciences Research, Ajman University, Ajman P.O. Box 346, United Arab Emirates; 10School of Pharmacy, The University of Jordan, Amman 11942, Jordan

**Keywords:** Mpox, pharmacists, UAE, knowledge, diagnose, management

## Abstract

Background: The WHO classified the mpox outbreak as a worldwide health emergency. Increasing the contribution of healthcare providers, such as pharmacists, can enhance preventive efforts. Assessing the knowledge and confidence levels of pharmacists in diagnosing and managing mpox cases can shape the response strategies necessary for the management of such outbreaks. Methods: This research employed a cross-sectional survey designed to assess the knowledge and preparedness of pharmacy students and pharmacists in the United Arab Emirates (UAE) regarding the mpox virus outbreak. Independent researchers evaluated the survey items to confirm the face and content validity of the developed survey. The final study’s survey was structured into three distinct sections, each addressing a specific area of interest. Data were analyzed using the IBM SPSS Statistics. Results: The 388 participants had a median age of 22.00 years (IQR = 5.00). The survey revealed that participants primarily relied on the WHO reports for mpox information (79.8%). The total knowledge scores (TK score) varied, ranging from −6 to 23 (median = 6.00), and symptom knowledge scores (SK score) ranged from −3 to 9 (median = 2.00). Older participants (*p*-value = 0.008) and females (*p*-value = 0.014) exhibited significantly higher TK scores. Only about 31.0% of participants expressed confidence in diagnosing mpox cases, and 34.6% expressed confidence in managing mpox cases. Nearly a quarter of the participants (24.5%) thought that getting vaccinated against COVID-19 led to contracting mpox more likely, whereas 45.7% believed that a previous infection with COVID-19 increases the risk of having mpox and its associated symptoms. Many respondents (38.7%) expressed their concern that mpox could emerge as the next major epidemic following COVID-19. Conclusion: Although pharmacists and pharmacy students in the UAE are aware of mpox, their knowledge and confidence levels in diagnosing and managing vary significantly. These findings suggest the need for targeted educational programs to enhance the understanding and preparedness of pharmacists to manage and prevent mpox cases.

## 1. Introduction

Mpox, a viral zoonotic infection from the *Orthopoxvirus* genus, was first discovered in 1958 when outbreaks occurred among research monkeys in Denmark. The virus then moved to humans, with the first confirmed case in a nine-month-old boy from the Democratic Republic of the Congo in 1970 [1]. Subsequently, mpox emerged in central, east, and west Africa [1,2].

Common symptoms of mpox often include a rash that may last for a duration of two to four weeks [2,3]. This could be accompanied by symptoms such as headache, fever, muscle ache, back pain, fatigue, and swollen glands. Individuals particularly vulnerable to experiencing more severe symptoms are pregnant women, children, and immunocompromised patients, such as those suffering from HIV disease [2].

The transmission of mpox from person to person occurs directly through close contact, including face-to-face and skin-to-skin, or indirectly via fomites [2,4]. The typical incubation period (the duration between the infection and the onset of symptoms) begins within 6 to 13 days, although the range can be 5 to 21 days [2,5].

Mpox outbreak has escalated to a global health emergency, according to the World Health Organization (WHO) [2]. In addition, the Centers for Disease Control and Prevention’s “2022–2023 Global Map & Case Count” reported over 97,200 confirmed cases of mpox across 118 locations in the recent multicounty outbreak, and particularly in the United Arab Emirates (UAE), there have been 16 confirmed cases of mpox [6].

Antiviral medications are available for treatment, whereas vaccination is the primary method of prevention [7]. As reported by the WHO, three vaccines (MVA-BN, LC16, and OrthopoxVac) originally developed for smallpox have been authorized for mpox as well; nevertheless, individuals who are at high risk should be considered for vaccination [2]. Moreover, preventive measures, such as increasing vaccination rates and raising public awareness, can be enhanced by activating the pivotal role of healthcare providers, who can provide education, promote vaccination, and contribute to the management of the disease [5].

Pharmacists play a growing role in the management of infectious diseases, particularly in providing patient education [8], ensuring medication adherence [9], and supporting vaccination efforts [10]. For instance, pharmacists have been actively involved in managing diseases such as COVID-19, where their responsibilities included disease prevention and infection control, proper storage and supply of medications, as well as providing patient care and support to healthcare professionals [11]. A review examining pharmacists’ involvement in disaster situations found that incorporating their new roles within healthcare systems can lead to more effective disaster response, as demonstrated during the COVID-19 pandemic [12].

The rising number of mpox cases highlights the critical role of healthcare providers, including pharmacists, in being equipped with prevention strategies, early identification of possible cases, and management measures following diagnosis [13]. Enhancing healthcare providers’ capacity for mpox recognition, diagnosis, and management is key to effective surveillance and improved patient outcomes [14]. However, a WHO report highlighted a significant obstacle in preventing the transmission of mpox, which is the low mpox knowledge, particularly among healthcare providers [14].

In light of the above, the primary objective of this study was to assess the knowledge and confidence levels of pharmacists and pharmacy students in diagnosing and managing mpox. This assessment can play a crucial role in guiding the response strategies required to effectively control the outbreak and draw future guidelines for the management of this and similar outbreaks.

## 2. Methods

### 2.1. Study Design and Participants

A descriptive cross-sectional study design was employed to evaluate the level of knowledge and preparedness of pharmacists and pharmacy students in the UAE regarding mpox. Participation was entirely voluntary and ensured anonymity to minimize any potential risk of bias.

### 2.2. Survey Development

An extensive review of the literature was performed to draft the initial version of this study’s survey [2,5,15,16,17]. Various sources were used to produce a diverse array of items that were in line with this study’s objectives. The literature review focused on mpox epidemiology, transmission, symptoms, incubation period, and preventive measures. The search was performed using PubMed, where keywords such as mpox, epidemiology, outbreak, knowledge, and public health emergency were utilized. Peer-reviewed articles published recently were included to ensure that the survey was developed with the most current and relevant scientific evidence. Subsequently, the research team ensured that the content was aligned with the main aims and objectives of this study. Redundant or unclear items were removed, and concepts were consolidated to improve clarity.

To establish the survey’s face and content validity, external reviewers were recruited. They analyzed the survey’s items for comprehension, relevance, and clarity of wording. The reviewers were chosen with a diverse range of expertise, including pharmacy practice, pharmacology, and public health. Furthermore, the survey was piloted with a sample of 37 participants whose responses were excluded from the final analysis. The purpose of this pilot study was to evaluate the survey’s comprehension, clarity, readability, and overall acceptability. Internal consistency was measured using Cronbach’s alpha, yielding a coefficient of 0.79.

The final survey was divided into three distinct sections, each targeting a specific aspect. The initial section consisted of 7 items aimed at assessing participants’ demographic characteristics. One of the questions in this first section asked participants about the number of workshops they attended, referring to continuing professional development, to gauge the overall educational engagement of pharmacists.

The second section covered eight sources from which participants obtained information about mpox, alongside 26 items evaluating their knowledge items.

The last section consisted of five items designed to evaluate participants’ self-reported ability to diagnose and manage possible mpox cases, along with their perceptions regarding the future implications of the disease. Asking pharmacists about diagnosing mpox in the current study did not imply that they perform formal diagnoses, as this remains the responsibility of physicians or other licensed healthcare professionals. Instead, the pharmacists’ role was to identify potential medication use and provide recommendations based on clinical guidelines and the patient’s reported symptoms or health status.

The 26 knowledge items comprised multiple answers with responses of true, false, or “I do not know”. Each correct answer received a score of 1, whereas incorrect answers were assigned −1. “I do not know” responses were assigned a value of 0. This approach yielded a total knowledge score (TK score) ranging from −26 (all incorrect) to 26 (all correct). Furthermore, a symptom knowledge score (SK score) was derived from a subset of 9 items specifically focused on mpox symptomology. Higher scores indicated a better awareness.

The research team decided to assign a score of 0 to the “I do not know” responses as this option reflects the participant’s lack of knowledge; hence, they do not actively provide an incorrect answer; this approach differentiates between a ‘lack of knowledge’ and the presence of ‘misinformation’ among the participants.

### 2.3. Survey Implementation

A multifaceted recruitment strategy was employed. This study’s survey link, preceded by a summarized overview of this study’s objectives and ethics committee approval, was distributed through social media platforms, mainly Facebook and Instagram. Specifically, relevant Facebook groups centered on pharmacy, pharmacy practice, and pharmacy students were used to effectively reach the potential participants. The email was also used to recruit potential participants. The email addresses were obtained through professional networks, academic institutions, and prior research collaborations, ensuring reaching individuals interested in pharmacy.

Google Forms enabled the survey administration, ensuring a user-friendly experience designed for completion within a 5 to 7 min timeframe.

### 2.4. Sample Size

The minimum required sample size was calculated to be 385, based on a 5% margin of error and a presumed response distribution of 50%. The calculation used the following formula: $n=\frac{4p(1-p)}{SE2}$, where *n* represents the sample size, *p* represents the expected prevalence, and SE represents the sampling error.

### 2.5. Statistical Analysis

After coding the participants’ responses, they were imported into a database using the Statistical Package for the Social Sciences (SPSS), Version 27.0 (IBM Corp., Armonk, New York, NY, USA). Categorical variables were represented using frequency and percentage, while continuous variables were represented using median and interquartile range (IQR), depending on data distribution. Data normality was assessed using the Shapiro–Wilk test, with a *p*-value ≥ 0.05 indicating a normally distributed continuous variable.

Screening of the factors (age, sex, marital status, education, and emirate) affecting the TK score among the study participants was conducted using the logistic regression analysis. For the simple logistic regression, any variable demonstrating significance on a single predictor level (*p* < 0.25) was included in the subsequent multiple logistic regression. The TK score was categorized as follows: 0 for “Low,” representing participants with a TK score below the median of 6, and 1 for “High”, indicating participants with a TK score of 6 or higher. Independent variables were selected after ensuring their independence, with tolerance values above 0.2 and Variance Inflation Factor (VIF) values below 5, confirming the absence of multicollinearity among the independent variables in regression analysis.

Multiple logistic regression was also used to screen the factors affecting the ability of the participant to diagnose and manage mpox cases.

### 2.6. Ethical Approval

Ethical approval for this study was obtained from the Research Ethics Committee at the University of Sharjah (Approval Number: REC-22-12-07).

## 3. Results

### 3.1. Demographic of the Recruited Participants

The 388 participants had a median age of 22.00 (IQR = 5.00) years. Females constituted about 79.0% of the participants (n = 306), and almost 80.0% of the participants were single (n = 309). Of the participants, 45.3% resided in Abu-Dhabi (n = 176), 33.2% in Sharjah (n = 129), and 9.5% in Ajman (n = 37). With regard to participants’ educational level, 39.7% held a bachelor’s degree in pharmacy (n = 154), 13.4% had a postgraduate degree (n = 52), and 45.1% of the participants were students (n = 175). Table 1 provides a detailed overview of the demographic of the recruited participants.

Regarding registered pharmacists (n = 213), 46 individuals reported working in academic fields, 93 in community pharmacies, 15 in government departments, 21 as employees in companies, and 38 reported other professional fields.

Out of the 175 pharmacy students, 10.4% were in their first year (n = 18), 20.8% were second-year students (n = 36), 17.4% were in their third year (n = 31), 32.8% were in their fourth year (n = 57), 15.8% were fifth-year students (n = 28), and only 2.7% (n = 5) were in their sixth year (Pharm.D. students).

Regarding attending workshops that covered various general topics, above one-quarter of the study participants (32.6%, n = 126) reported not attending any general workshop per year, 16.7% attended one workshop (n = 65), 21.5% attended two workshops (n = 84), 19.3% attended three workshops (n = 75), 5.7% attended four workshops (n = 22), and 4.2% attended five workshops or more (n = 16).

The survey revealed that participants primarily relied on the WHO reports for mpox information (Figure 1), with a utilization rate of 79.8%. Followed closely are published research (74.7%) and government awareness campaigns (68.5%).

### 3.2. Mpox Virus Knowledge

Assessing the participants’ knowledge levels regarding the mpox virus revealed that the most correctly answered statement was “*Fever can occur as a symptom of monkeypox*”, with 84.8% of the participants answering it correctly. Furthermore, 73.7% of the participants correctly answered that “*The rash may appear on the face, feet, and hands*”, 72.2% correctly identified “*Muscle aches are among the symptoms of monkeypox*”, 71.1% correctly identified “*Headaches can occur as a symptom of monkeypox*”, and 70.4% correctly identified that “*A rash from monkeypox may persist for 2–3 weeks*”. Besides the five previously mentioned items, the following five items were answered correctly by more than 55.0% of the participants: “*Animal-to-human transmission of monkeypox is possible*”, “*Personal items can be a transmission route for monkeypox*”, “*Contact with contaminated surfaces can lead to infection, especially if the person has cuts or abrasions*”, “*Swollen lymph nodes are among the symptoms of monkeypox*”, and “*The incubation period for monkeypox ranges from 6 to 13 days*” (70.1%, 65.2%, 63.7%, 57.5%, 56.2%, respectively). On the other hand, the least correctly answered statement was “*After close contact with a monkeypox case, you must monitor yourself for symptoms over the next 14 days*”, with only 5.2% (n = 20) of the participants identifying it as incorrect since the accurate timeframe should be 21 days instead of 14 days (Figure 2).

The range of total knowledge scores (TK score) among participants varied from −6 to 23. These scores had a median of 6.00 (IQR = 6.00) on a scale from −26 (lowest possible score) to 26 (highest possible score). Surprisingly, none of the participants answered all knowledge items correctly (Figure 3).

The participants’ symptom knowledge scores (SK score), measured on a scale from −9 to 9, ranged from −3 to 9, with a median of 2.00 (IQR = 3.00). It is worth mentioning that only eight participants (2.06%) answered all the symptom knowledge questions correctly (Figure 3).

Multiple logistic regression revealed that the TK score was significantly affected by the participant’s age and gender (Table 2). Older participants and females had higher TK scores (*p*-value = 0.008 and 0.014, respectively).

### 3.3. The Participants’ Levels of Confidence in Diagnosing and Managing Mpox Cases

Regarding participants’ confidence in their ability to diagnose mpox cases based on their current knowledge and skills, 30.7% responded affirmatively. However, a slightly higher percentage of participants (34.6%) expressed confidence in managing mpox cases (Figure 4).

Multiple logistic regression analyses of factors affecting the capability to diagnose mpox cases among the study participants highlighted that the place of work (indicated as an emirate variable) was the only significant variable (Table 3). However, none of the variables significantly affected the ability to manage mpox cases.

### 3.4. Perception of Mpox Virus

The survey identified misperceptions about mpox. Nearly a quarter (24.5%) of the participants reported that COVID-19 vaccination increases the risk of contracting mpox and facing severe symptoms, whereas 45.7% thought that prior COVID-19 infection or long-COVID-19 raises susceptibility and complication risk for mpox. Furthermore, 38.7% of participants expressed concern that mpox could evolve into the next epidemic (Figure 5).

## 4. Discussion

The present study is the first of its kind in the UAE that examined the knowledge of pharmacists and future pharmacists on their knowledge to identify and potentially manage mpox cases within the UAE context. This study’s results indicated general awareness of the mpox virus among the participants, yet this study uncovered certain knowledge and confidence gaps that could potentially impact the diagnosis and management of mpox cases negatively.

The WHO reports were the most used source of information regarding mpox among this study’s participants, followed by published research and government awareness campaigns. The dependence on these trustworthy sources is promising. However, the diversity in knowledge scores among the participants highlights the need for more targeted and comprehensive educational initiatives. For example, incorporating up-to-date information about mpox in training programs can bridge the knowledge gaps.

The symptom that most participants commonly agreed upon was fever, followed by muscle aches and headaches. These common symptoms are all supported by published articles confirming the main symptoms of the mpox virus, including fever, headache, fatigue, muscle pain, generalized body aches, lymph node swelling, and skin lesions [18].

Most of the participants correctly identified that mpox is not a bacterial zoonotic infection. Supporting the consistent classification of mpox as a viral infection in the previously published literature [19].

Despite over two-thirds of pharmacists recognizing the term “monkeypox” and its associated symptoms, their TK scores varied widely, and none of the pharmacists answered all items correctly. This indicates substantial room for improvement in the detailed understanding of the disease.

The findings of the present study align with recent research that indicated an acceptable level of mpox knowledge. In Egypt, healthcare providers exhibited a low level of knowledge about mpox [20]. In Kuwait, there was a demonstrated lack of knowledge about mpox, particularly concerning the transmission of the virus and the non-cutaneous symptoms of the disease [5]. A cross-sectional study was conducted to evaluate mpox knowledge among Vietnamese university students. The results indicated that students had an acceptable level of understanding regarding the prevalence, primary modes of transmission, incubation period, symptoms, and prevention of mpox. Participants were assessed on 21 knowledge items, with pharmacy students achieving an average knowledge score of 12.139 (SD = 3.545) [21].

The current study identified age and gender as significant factors affecting the TK score, with older participants and females scoring higher. This could be due to factors such as the experience or a greater focus on empathy and patient interaction among female pharmacists, which enhances knowledge retention [22,23]. On the other hand, the educational level did not significantly affect the TK scores, indicating a potential need for current curricula updates to better address emerging infectious diseases like mpox.

The logistic regression analysis showed that the place of work (emirate variable) was the only factor significantly associated with the ability to diagnose mpox cases, suggesting regional differences in healthcare infrastructure or training. Future research could further explore the previous relationship and other factors, such as specialized training or resource availability.

The current study assessed the pharmacists’ perceptions of the mpox virus. Notably, 38.7% expressed concern about mpox becoming a future epidemic, which could lead to increased vigilance in monitoring cases and educating patients about prevention measures. Additionally, about a quarter of the participants believed that people vaccinated against COVID-19 are at higher risk of experiencing severe symptoms from mpox. This perception could drive pharmacists to enhance their educational outreach and patient counseling.

Further research is recommended to investigate the concerns of healthcare professionals and the general public about a potential mpox outbreak. Additionally, measures should be taken to raise awareness of how to prevent such an outbreak. In addition, future research should explore the differentiation between active practitioners and non-practicing individuals, such as those holding a PhD. This stratification could enhance the analysis and provide more targeted insights regarding the knowledge and practices related to mpox management among pharmacists.

One of this study’s limitations lies in its sampling methods, the likelihood that certain individuals are more likely to complete this study’s survey than others. Potential participants decide independently whether to take part, leading to “self-selection bias”. Self-selection bias may limit the generalizability of this study’s findings, potentially affecting the accuracy of conclusions about pharmacists’ perceptions and knowledge of mpox. To address this, the survey was distributed through various professional networks to capture a broader range of perspectives.

Another limitation of this study is the low proportion of males to females among the participants, which may affect the generalizability of the findings. Furthermore, factor analysis was not conducted to further validate the questionnaire’s construct validity. Although students from all academic years participated in the current study, allowing for a range of perspectives, the students in the early years may have had limited formal education on the subject. The extent to which the topic is covered in the curriculum and the manner of its instruction may vary, which presents a potential limitation of this study.

## 5. Conclusions

Pharmacists and pharmacy students in the UAE were aware of mpox; however, their knowledge and confidence levels in diagnosing and managing varied significantly. The present study underscores the urgent need for improved educational programs to enhance the understanding and confidence of pharmacists and pharmacy students in the UAE regarding mpox.

Addressing the identified knowledge gaps is crucial for pharmacists to play a more effective role in preventing and managing mpox, ultimately supporting public health efforts in controlling potential outbreaks.

## Figures and Tables

**Figure 1 healthcare-12-02295-f001:**
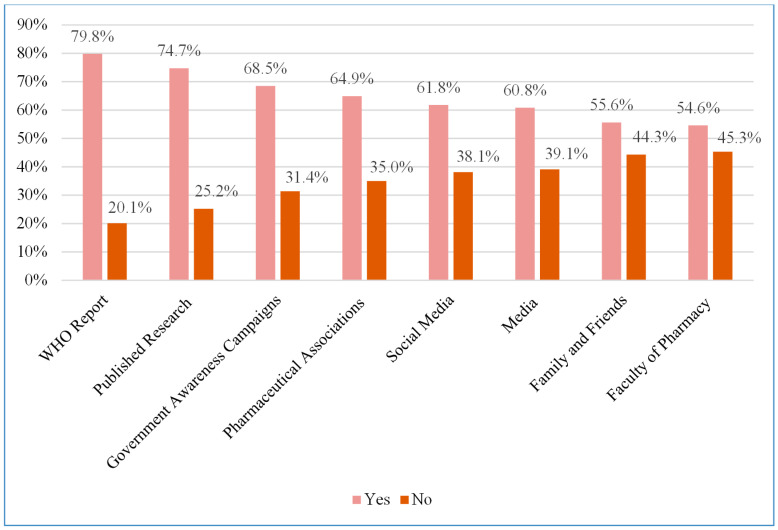
The sources of information utilized by the study participants regarding mpox (n = 388).

**Figure 2 healthcare-12-02295-f002:**
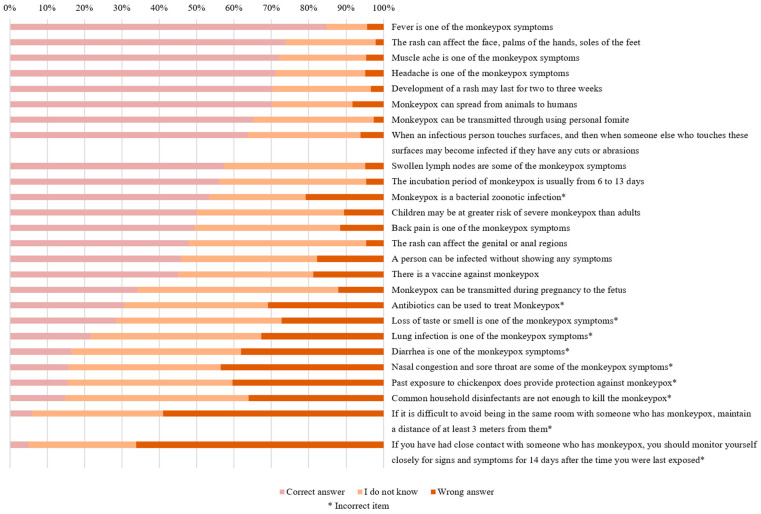
Responses to the 26 knowledge items among the study participants (n = 388).

**Figure 3 healthcare-12-02295-f003:**
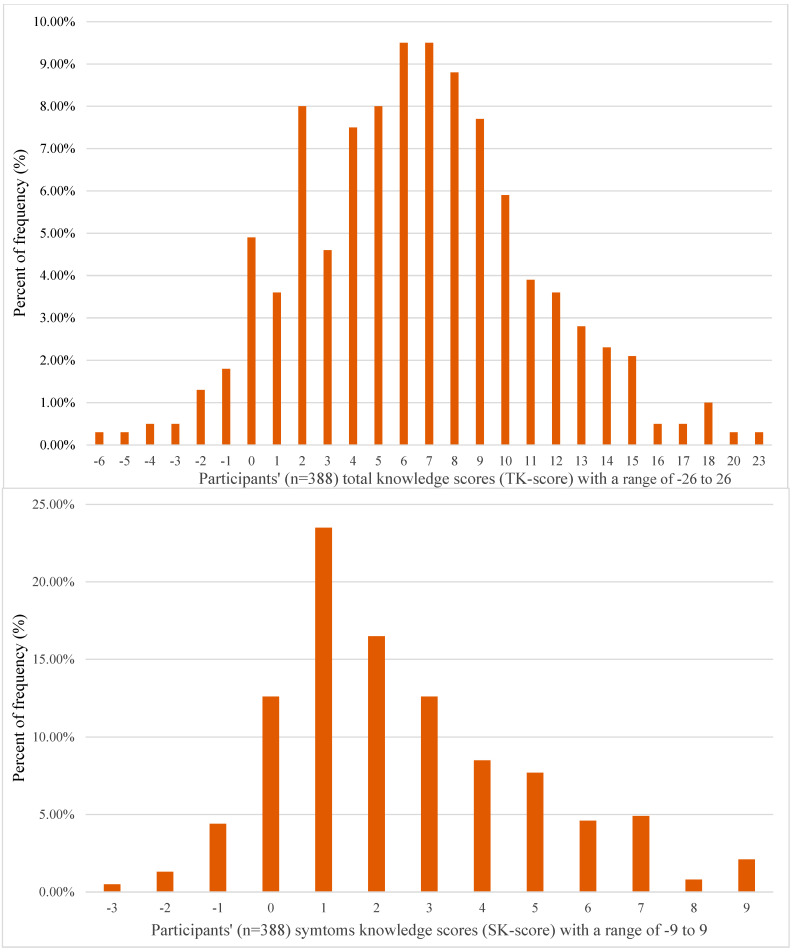
Study participants’ total knowledge and symptom knowledge scores (n = 388).

**Figure 4 healthcare-12-02295-f004:**
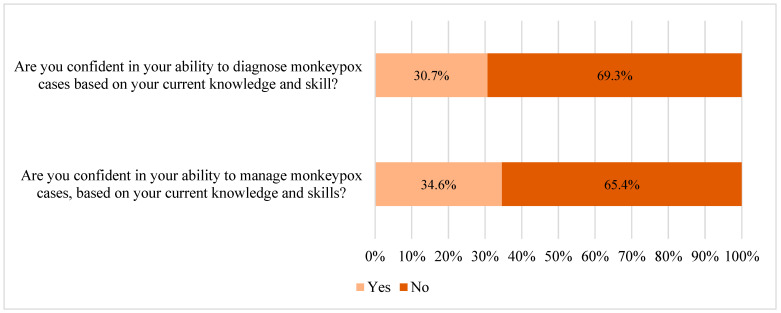
Assessment of the confidence level in diagnosing and managing mpox cases among the participants (n = 388).

**Figure 5 healthcare-12-02295-f005:**
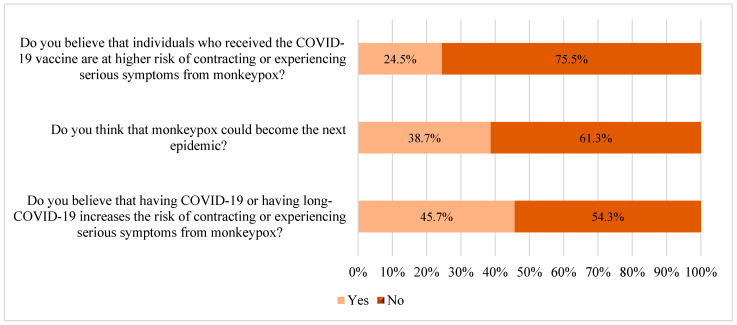
Perception of mpox virus among the study participants (n = 388).

**Table 1 healthcare-12-02295-t001:** The demographic of the recruited pharmacists and pharmacy students (n = 388).

Parameter	n (%)
**Sex**	
Male	82 (21.1)
Female	306 (78.9)
**Marital Status**	
Single	309 (79.6)
Married	76 (19.6)
Divorced	2 (0.5)
Widowed	1 (0.3)
**Emirate**	
Abu-Dhabi	176 (45.3)
Ajman	37 (9.5)
Dubai	34 (8.8)
Fujairah	2 (0.5)
Ras Al-Khaimah	6 (1.5)
Sharjah	129 (33.2)
Umm Al-Quwain	4 (1.0)
**Education**	
I am currently pursuing my studies	175 (45.1)
Diploma degree	7 (1.8)
Bachelor’s degree	154 (39.7)
Master’s degree	35 (9.0)
PhD degree	17 (4.4)

**Table 2 healthcare-12-02295-t002:** Factors affecting the total knowledge score (TK score) among the study participants (n = 388).

Parameter	TK Score			
	OR	*p*-Value ^#^	OR	*p*-Value ^$^
**Age**	1.073	<0.001	1.074	0.008 *
**Sex**				
Male	Reference			
Female	1.398	0.180	1.946	0.014 *
**Marital status**				
Single/Divorced/Widowed	Reference			
Married	2.400	0.002	1.014	0.975
**Education**				
<Bachelor’s Degree	Reference			
≥Bachelor’s Degree	0.865	0.484	1.232	0.377
**Emirate**				
Abu-Dhabi	Reference			
Others	1.745	0.007	----	----

^#^: using simple logistic regression; ^$^: using multiple logistic regression; * significant at 0.05 significance level.

**Table 3 healthcare-12-02295-t003:** Factors affecting the ability to diagnose mpox cases among the study participants (n = 388).

Parameter	Ability to Diagnose Mpox Cases			
	OR	*p*-Value ^#^	OR	*p*-Value ^$^
**Age**	1.018	0.216	0.989	0.634
**Sex**				
Male	Reference			
Female	0.926	0.774	----	----
**Marital status**				
Single/Divorced/Widowed	Reference			
Married	1.660	0.057	1.617	0.244
**Education**				
<Bachelor’s Degree	Reference			
≥Bachelor’s Degree	1.367	0.161	1.218	0.439
**Emirate**				
Abu-Dhabi	Reference			
Others	0.542	0.006	0.539	0.007 *

^#^: using simple logistic regression; ^$^: using multiple logistic regression; * significant at 0.05 significance level.

## Data Availability

Data are available on request due to privacy or ethical restrictions.

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
