# Peer review of "Knowledge Gap or Prepared Force? Exploring United Arab Emirates Pharmacy Students and Pharmacists’ Monkeypox Readiness"

_healthcare, 2024, doi:10.3390/healthcare12222295_

Round 1
Reviewer 1 Report
Comments and Suggestions for Authors
Overall, the paper was presented clearly. Some comments listed below:
In the results section, lines 141-144 - the percentages does not add to 100%.
Figure 1 - "Educational Attainment" could be written as only "Education" or "Educational Background"
Resolution of Figure 2 is low. The words are hard to read.
Figure 3: The word "participants" is misspelled on both X-axes.
In table 2, Row "Education" - the p value is not aligned the same as the other p-values in the table.
In table 3, p-value numbers are not aligned the same in the table.
• With the increase in Mpox cases recently, this article presents relevant information to the healthcare field. It showcases the current knowledge among healthcare professionals which is critical to developing a response system to disease surveillance.
• The methodology presented in the paper is clear. However, the percentage of male to females in the study is low. Also, the marital status may not be crucial to support the results from the study.
• The references are appropriate for the study.
• There are a few typos/edits needed for the figures that have been listed in my previous review of the paper.
• Overall, the results presented in this paper support the methodology. The conclusions derived from this study would be helpful to understand the current knowledge surrounding Mpox infections.
Overall, the quality of English language is good. Minor editing to texts and alignment of values in tables is required and is listed in the comments above.
Author Response
Comment
Overall, the paper was presented clearly. Some comments listed below:
Response
Thank you. Your time and effort in reviewing the manuscript are highly appreciated.
Comment
In the results section, lines 141-144 - the percentages does not add to 100%.
Response
The percentages now add to 100% (32.6%, 16.7%, 21.5%, 19.3%, 5.7%, & 4.2%)
Regarding attending workshops that covered various general topics, above one-quarter of the study participants (32.6%, n=126) reported not attending any general workshop per year, 16.7% attended one workshop (n=65), 21.5% attended two work-shops (n=84), 19.3% attended three workshops (n=75), 5.7% attended four workshops (n=22), and 4.2% attended 5 workshops or more (n=16).
Comment
Figure 1 - "Educational Attainment" could be written as only "Education" or "Educational Background"
Response
Done. "Educational Attainment" has been changed to "Education".
Comment
Resolution of Figure 2 is low. The words are hard to read.
Response
Figure 2 with a good resolution will be submitted to the journal, within this round of review.
Comment
Figure 3: The word "participants" is misspelled on both X-axes.
Response
Thank you for pointing out this typo. The word has been edited and written correctly.
Comment
In table 2, Row "Education" - the p value is not aligned the same as the other p-values in the table.
Response
Done. It was edited to align with the other p-values.
Comment
In table 3, p-value numbers are not aligned the same in the table.
Response
Thank you for bringing our attention to this point. The p-values are now aligned to each other
Comment
- With the increase in Mpox cases recently, this article presents relevant information to the healthcare field. It showcases the current knowledge among healthcare professionals which is critical to developing a response system to disease surveillance.
- The methodology presented in the paper is clear. However, the percentage of male to females in the study is low. Also, the marital status may not be crucial to support the results from the study.
- The references are appropriate for the study.
- There are a few typos/edits needed for the figures that have been listed in my previous review of the paper.
- Overall, the results presented in this paper support the methodology. The conclusions derived from this study would be helpful to understand the current knowledge surrounding Mpox infections
Response
Thank you for your valuable comments and insightful feedback.
Regarding the percentage of males to females, it was added as a limitation of the study as follows: "Another limitation of the study is the low proportion of males to females among the participants, which may affect the generalizability of the findings."
Reviewer 2 Report
Comments and Suggestions for Authors
The topic is interesting and under-researched, but certain revisions are necessary, and there is room for improvement in the paper.
1) At the very beginning, the authors are listed without affiliations.
2) There are many typos, even in the abstract where "UAW" is mentioned instead of "UAE," and throughout the paper, there are instances of bolded text where it shouldn’t be, double periods, incomplete sentences (e.g., line 190), and similar issues.
3) In the introduction and discussion, only monkeypox is discussed, with no mention of previous research on the knowledge of pharmacists (as well as healthcare workers and the general population) on this topic. In the past two years, several studies have been published that specifically examined knowledge about this virus.
4) When calculating the sample size, only the minimum number of participants needed is mentioned, without explaining how this number was determined.
5) In the survey development section, an "extensive review of the literature" is mentioned, but it is not explained what was searched for or included in the analysis.
6) The survey development section should also describe the final questionnaire and the scoring method (a brief mention is made in the statistical analysis, but this should be moved to the methods section and expanded upon).
7) It is mentioned that face and content validity were conducted and that questionnaire is divided into several sections. Was the reliability of the questionnaire calculated, and was a factor analysis performed?
8) Figure 2 is entirely blurry, and the questions cannot be read at all.
9) In the results section, line 201, it is mentioned that the place of work is the only significant variable, but it is not explained why or how it influences the possibility of diagnosis.
10) A significant number of respondents are students (45%). It is not mentioned which year of study they are in. If they are at the beginning of their studies, how informed can they be on the topic? There is no mention of whether this topic is covered in the curriculum, to what extent, and in what manner. If these are students in the early years who have not yet encountered this topic in their education, is this one of the study's limitations?
Comments on the Quality of English LanguageThe style, grammar, and typos need some minor corrections in certain places.
Author Response
Comment
The topic is interesting and under-researched, but certain revisions are necessary, and there is room for improvement in the paper.
Response
Thank you. Your time and effort in reviewing the manuscript are highly appreciated.
Comment
1) At the very beginning, the authors are listed without affiliations.
Response
The authors' affiliations have been added.
Comment
2) There are many typos, even in the abstract where "UAW" is mentioned instead of "UAE," and throughout the paper, there are instances of bolded text where it shouldn’t be, double periods, incomplete sentences (e.g., line 190), and similar issues.
Response
Thank you for your valuable feedback.
We have carefully revised the entire manuscript to correct these errors and ensure clarity and consistency throughout. Specifically, the typo in the abstract and the incomplete sentence on line 190 have been addressed.
Comment
3) In the introduction and discussion, only monkeypox is discussed, with no mention of previous research on the knowledge of pharmacists (as well as healthcare workers and the general population) on this topic. In the past two years, several studies have been published that specifically examined knowledge about this virus.
Response
The following paragraph was added:
"The findings of the present study align with recent research that indicated an ac-ceptable level of monkeypox knowledge. In Egypt, healthcare providers exhibited a low level of knowledge about monkeypox [18]. In Kuwait, there was a demonstrated lack of knowledge about monkeypox, particularly concerning the transmission of the virus and the non-cutaneous symptoms of the disease [5]. A cross-sectional study was conducted to evaluate monkeypox knowledge among Vietnamese university students. The results indicated that students had an acceptable level of understanding regarding the preva-lence, primary modes of transmission, incubation period, symptoms, and prevention of monkeypox. Participants were assessed on 21 knowledge items, with pharmacy stu-dents achieving an average knowledge score of 12.139 (SD=3.545) [19]."
Comment
4) When calculating the sample size, only the minimum number of participants needed is mentioned, without explaining how this number was determined.
Response
The following paragraph was added to the sample size:
"The minimum required sample size was calculated to be 385, based on a 5% margin of error and a presumed response distribution of 50%. The calculation used the formula: , where n represents the sample size, p represents the expected prevalence, and SE represents the sampling error."
Comment
5) In the survey development section, an "extensive review of the literature" is mentioned, but it is not explained what was searched for or included in the analysis.
Response
The following paragraph was added:
"The literature review focused on monkeypox epidemiology, transmission, symptoms, incubation period, and preventive measures. The search was performed using PubMed, where keywords such as monkeypox, epidemiology, outbreak, knowledge, and public health emergency were utilized. Peer-reviewed articles published recently were included to ensure that the survey was developed with the most current and relevant scientific evidence."
Comment
6) The survey development section should also describe the final questionnaire and the scoring method (a brief mention is made in the statistical analysis, but this should be moved to the methods section and expanded upon).
Response
The following paragraph was added to the survey development"
"The 26 knowledge items comprised multiple answers with responses of true, false, or "I do not know." Each correct answer received a score of 1, whereas incorrect answers were assigned -1. "I do not know" responses were assigned a value of 0. This approach yielded a total knowledge score (TK-score) ranging from -26 (all incorrect) to 26 (all correct). Furthermore, a symptoms knowledge score (SK-score) was derived from a subset of 9 items specifically focused on monkeypox symptomology. Higher scores in-dicated a better awareness."
Comment
7) It is mentioned that face and content validity were conducted and that questionnaire is divided into several sections. Was the reliability of the questionnaire calculated, and was a factor analysis performed?
Response
The reliability was calculated; however, the factor analysis was not performed.
1. The following paragraph was added to clarify that the Cronbach alpha was calculated
"To establish the survey's face and content validity, external reviewers were recruited. They analyzed the survey's items for comprehension, relevance, and clarity of wording. Furthermore, the survey was piloted with a sample of 37 participants, whose responses were excluded from the final analysis. The purpose of this pilot study was to evaluate the survey’s comprehension, clarity, readability, and overall acceptability. Internal consistency was measured using Cronbach’s alpha, yielding a coefficient of 0.79."
2. Not performing factor analysis was added to the study's limitation, as follows:
"Furthermore, factor analysis was not conducted to further validate the questionnaire's construct validity."
Comment
8) Figure 2 is entirely blurry, and the questions cannot be read at all.
Response
Figure 2 with a good resolution will be re-submitted to the journal, within this round of review.
Comment
9) In the results section, line 201, it is mentioned that the place of work is the only significant variable, but it is not explained why or how it influences the possibility of diagnosis.
Response
The following paragraph was added to the discussion part:
"The logistic regression analysis showed that the place of work (Emirate variable) was the only factor significantly associated with the ability to diagnose monkeypox cases, suggesting regional differences in healthcare infrastructure or training. Future research could further explore the previous relationship, and other factors such as specialized training or resource availability."
Comment
10) A significant number of respondents are students (45%). It is not mentioned which year of study they are in. If they are at the beginning of their studies, how informed can they be on the topic? There is no mention of whether this topic is covered in the curriculum, to what extent, and in what manner. If these are students in the early years who have not yet encountered this topic in their education, is this one of the study's limitations?
Response
Thank you for your insightful comment.
The current year of study was indeed collected, as has been added to the paper in the demographic data, as follows:
"Out of the 175 pharmacy students, 10.4% were in their first year (n=18), 20.8% were second-year students (n=36), 17.4% were in their third year (n=31), 32.8% were in their fourth year (n=57), 15.8% were fifth-year students (n=28), and only 2.7% (n=5) were in their sixth year (Pharm.D. students)."
Moreover, the current paragraph was added to the limitation:
"Although, students from all academic years participated in the current study, allowing for a range of perspectives; the students in the early years may have had limited formal education on the subject. The extent to which the topic is covered in the curriculum and the manner of its instruction may vary, which presents a potential limitation of the study."
Reviewer 3 Report
Comments and Suggestions for Authors
This manuscript describes a cross-sectional survey study of pharmacy students and pharmacy students in the UAE concerning knowledge and perceptions of monkeypox. The topic has gained in interest in the last few years globally and pharmacists can play a role in disease prevention and management. However, this role is not made clear in the manuscript itself and there are questions regarding the methodology which limit the impactfulness of the study itself. Specific comments are provided:
Abstract
1) Follow formatting instructions which notes that there should be no section headings within the abstract.
2) Lines 14-15: The software is now just referred to as “IBM SPSS Statistics”.
3) Line 20: This is also noted in the methods of the main text, but pharmacists often do not diagnose, unless they have that authority within the UAE.
Introduction
4) Lines 38-39: This statement is too general and not needed. The following sentences contain more important specifics about monkeypox.
5) Lines 58-59: Not sure what this sentence means. The preventive measures should be specified. For example, is this suggesting that health care professionals can enhance vaccination rates?
Methods
6) Lines 68-69: To what degree will pharmacists be diagnosing monkeypox? Is this a typical (or is diagnosis in general within the scope of practice) for pharmacists internationally? Additionally, the Introduction should reference the degree to which pharmacists are currently involved in the management of conditions similar to monkeypox. This is important missing content that is needed to place the current study within the broader literature base.
7) Lines 81-82: Explain how a thematic analysis is used in this fashion as it is typically a formal qualitative data analysis technique.
8) Line 85: What were the expertise areas of the reviewers? This is important for actually establishing face and content validity and because the knowledge-based questions developed were a core part of the survey content.
9) Line 96: Need more detail here. What social media platforms were used? Were certain groups targeted (e.g., Facebook groups)? How were e-mails obtained to actually send the survey out?
10) Line 107: Given the data distribution, was the mean the most appropriate presentation of the data or was a median more appropriate?
11) Lines 109-110: Why not count “I don’t know” as incorrect? The respondent if they select this option does not know the answer and could not provide the correct answer if asked this question by a patient or other health care professional.
12) Lines 115-118: Specify the variables tested for potential inclusion in the final logistic regression model.
Results
13) Line 130: Do the authors have any sense of how many individuals were potentially reached by the survey distribution methods used?
14) Line 130: Were any participants excluded (e.g., for incomplete surveys)? Also, years in practice would have been a more useful variable compared to age (even if these variables are often correlated).
15) Lines 131-132: Why did marital status matter? What impact was this variable hypothesized to have on the study topic?
16) Lines 138-140: Why not ask students about their employment status? Many students may actually be working in a pharmacy (even if this is not as a pharmacist) unless this is not allowed in the UAE. Also, why not limit the survey to practicing pharmacists who would actually be responsible for managing monkeypox? Without this limit/exclusion criterion, it is likely that some proportion of respondents have a pharmacy degree (or the 4.4% with a PhD) who do not practice and would be less likely to have obtained the information being asked about in the survey.
17) Lines 141-144: What is a “workshop” in this context? Are pharmacists required to attend these workshops? Did this specifically focus on monkeypox? How does this question relate to the study topic?
18) Line 171: Why is a mean reported when a median was used to stratify TK scores into a binary (high/low) variable?
19) Lines 173-174 and 177-178: Why report frequencies of total scores rather than just a median, which is a more effective and reader-friendly way to present the same data?
Discussion
20) Line 226: Is “disproved” the right term to use? That typically refers to scientific evidence rather than survey responses.
21) Lines 228-233: This was not identified in the results as a key finding, yet it is presented here as being highly impactful. This discrepancy needs to be rectified.
22) Lines 240-241: Need a citation to support the statement that empathy enhances knowledge retention and that females exhibit more empathy than their male colleagues particularly within the health care setting.
23) Lines 253-254: Impact it in what way? Interpret the results rather than just restating them.
24) Lines 258-260: How do you think these limitations could have potentially impacted study findings? Was anything done to try and mitigate these limitations?
Comments on the Quality of English LanguageOnly minor grammatical/language concerns to address. Can be easily addressed by a read-through from a native English speaker.
Author Response
Comment
This manuscript describes a cross-sectional survey study of pharmacy students and pharmacy students in the UAE concerning knowledge and perceptions of monkeypox. The topic has gained in interest in the last few years globally and pharmacists can play a role in disease prevention and management. However, this role is not made clear in the manuscript itself and there are questions regarding the methodology which limit the impactfulness of the study itself. Specific comments are provided:
Response
Thank you. Your time and effort in reviewing the manuscript are highly appreciated.
Abstract
Comment
1) Follow formatting instructions which notes that there should be no section headings within the abstract.
Response
Done. The section headings in the abstract have been removed
Comment
2) Lines 14-15: The software is now just referred to as “IBM SPSS Statistics”.
Response
"Statistical Package for the Social Sciences" has been changed to "IBM SPSS Statistics"
Comment
3) Line 20: This is also noted in the methods of the main text, but pharmacists often do not diagnose, unless they have that authority within the UAE.
Response
Thank you for your insightful comment.
We agree that pharmacists typically do not diagnose, as diagnosis is generally reserved for physicians or other authorized healthcare professionals.
In our study, we did not imply that pharmacists were performing formal diagnoses. Rather, their role was to identify potential medication-related issues and offer recommendations based on clinical guidelines and patient symptoms.
The methods section has been revised as follows to clarify this distinction
"Asking pharmacists about diagnosing monkeypox in the current study did not imply that they perform formal diagnoses, as this remains the responsibility of physicians or other licensed healthcare professionals. Instead, the pharmacists’ role was to identify potential medication use, and provide recommendations based on clinical guidelines and the patient's reported symptoms or health status."
Introduction
Comment
4) Lines 38-39: This statement is too general and not needed. The following sentences contain more important specifics about monkeypox.
Response
The sentence "Monkeypox has the potential to induce several signs and symptoms, with infection severity varying among infected individuals" has been deleted
Comment
5) Lines 58-59: Not sure what this sentence means. The preventive measures should be specified. For example, is this suggesting that health care professionals can enhance vaccination rates?
Response
The following sentence "Moreover, preventive measures can be enhanced by activating the pivotal role of healthcare providers"
has been changed to:
"Moreover, preventive measures such as increasing vaccination rates and raising public awareness, can be enhanced by activating the pivotal role of healthcare providers, who can provide education, promote vaccination, and contribute to the management of the disease"
Methods
Comment
6) Lines 68-69: To what degree will pharmacists be diagnosing monkeypox? Is this a typical (or is diagnosis in general within the scope of practice) for pharmacists internationally? Additionally, the Introduction should reference the degree to which pharmacists are currently involved in the management of conditions similar to monkeypox. This is important missing content that is needed to place the current study within the broader literature base.
Response
- To clarify the diagnosing point, the following paragraph was added:
"Asking pharmacists about diagnosing monkeypox in the current study did not imply that they perform formal diagnoses, as this remains the responsibility of physicians or other licensed healthcare professionals. Instead, the pharmacists’ role was to identify potential medication use, and provide recommendations based on clinical guidelines and the patient's reported symptoms or health status."
- The following paragraph was added to the introduction section:
"Pharmacists play a growing role in the management of infectious diseases, particularly in providing patient education [8], ensuring medication adherence [9], and sup-porting vaccination efforts [10]. For instance, pharmacists have been actively involved in managing diseases such as COVID-19, where their responsibilities included disease prevention and infection control, proper storage and supply of medications, as well as providing patient care and support to healthcare professionals [11]. A review examining pharmacists' involvement in disaster situations found that incorporating their new roles within healthcare systems can lead to more effective disaster response, as demon-strated during the COVID-19 pandemic [12]."
Comment
7) Lines 81-82: Explain how a thematic analysis is used in this fashion as it is typically a formal qualitative data analysis technique.
Response
The following sentence "Subsequently, the research team conducted a thematic analysis to refine the survey content. "
To: "Subsequently, the research team ensured that the content was aligned with the main aims and objectives of the study."
Comment
8) Line 85: What were the expertise areas of the reviewers? This is important for actually establishing face and content validity and because the knowledge-based questions developed were a core part of the survey content.
Response
The following paragraph was added: "The reviewers were chosen with a diverse range of expertise, including pharmacy practice, pharmacology, and public health"
Comment
9) Line 96: Need more detail here. What social media platforms were used? Were certain groups targeted (e.g., Facebook groups)? How were e-mails obtained to actually send the survey out?
Response
To clarify this point, the following paragraph was added to the "Survey Implementation"
"A multifaceted recruitment strategy was employed. The study's survey link, pre-ceded by a summarized overview of the study's objectives and ethics committee approval, was distributed through social media platforms; mainly Facebook and Insta-gram. Specifically, relevant Facebook groups centered on pharmacy, pharmacy practice, and pharmacy students were used to effectively reach the potential participants. The email was also used to recruit potential participants. The email addresses were obtained through professional networks, academic institutions, and prior research collaborations, ensuring reaching individuals interested in pharmacy."
Comment
10) Line 107: Given the data distribution, was the mean the most appropriate presentation of the data or was a median more appropriate?
Response
- The following sentence "Categorical variables were represented using frequency and percentage, while continuous variables were represented using mean and standard deviation."
Was edited as follows:
"Categorical variables were represented using frequency and percentage, while continuous variables were represented using median and interquartile range (IQR), depending on data distribution. Data normality was assessed using the Shapiro-Wilk test, with a p-value ≥ 0.05 indicating a normally distributed continuous variable."
- Additionally, the mean was replaced with the median in three places throughout the manuscript.
Comment
11) Lines 109-110: Why not count “I don’t know” as incorrect? The respondent if they select this option does not know the answer and could not provide the correct answer if asked this question by a patient or other health care professional.
Response
We chose to assign a score of 0 to the “I do not know” responses because this option reflects neutral responses; where the participant acknowledges their lack of knowledge, however; they do not actively provide an incorrect answer.
This approach helps to differentiate between a lack of knowledge and the presence of misinformation. In other words, by assigning a score of -1 to incorrect answers, we aim to account for participants who possess and express incorrect knowledge, which could have a more detrimental effect in clinical practice compared to simply not knowing the answer.
this was clarified in the methods section as follows:
"The research team decided to assign a score of 0 to the “I do not know” responses as this option reflects the participant's lack of knowledge, hence they do not actively provide an incorrect answer, this approach differentiates between 'a lack of knowledge' and the presence of 'misinformation' among the participants.
Comment
12) Lines 115-118: Specify the variables tested for potential inclusion in the final logistic regression model.
Response
The sentence has been edited as follows: "Screening of the factors (age, sex, marital status, education, and emirate) affecting the TK-score among the study participants was conducted using the logistic regression analysis."
Results
Comment
13) Line 130: Do the authors have any sense of how many individuals were potentially reached by the survey distribution methods used?
Response
Thank you for your question.
Unfortunately, we do not have specific data regarding the number of individuals potentially reached by the survey distribution methods used.
Comment
14) Line 130: Were any participants excluded (e.g., for incomplete surveys)? Also, years in practice would have been a more useful variable compared to age (even if these variables are often correlated).
Response
- No responses were excluded from the analysis.
- Years of experience could only be applicable to pharmacists who have graduated, whereas our study included both practicing pharmacists and students.
Nonetheless, we acknowledge the importance of this variable and have addressed it in the discussion. Specifically, "The current study identified age and gender as significant factors affecting the TK-score, with older participants and females scoring higher. This could be due to factors such as the experience ……"
Comment
15) Lines 131-132: Why did marital status matter? What impact was this variable hypothesized to have on the study topic?
Response
Thank you for the insightful comment.
Marital status was included in the study as a demographic variable, as it can potentially reflect differences in responsibilities, stress levels, and social interactions that might also influence knowledge or preparedness for handling health-related situations, including emerging infectious diseases. Marital status was hypothesized to possibly correlate with participants' confidence and access to information, which may affect their ability to diagnose and manage monkeypox.
Comment
16) Lines 138-140: Why not ask students about their employment status? Many students may actually be working in a pharmacy (even if this is not as a pharmacist) unless this is not allowed in the UAE. Also, why not limit the survey to practicing pharmacists who would actually be responsible for managing monkeypox? Without this limit/exclusion criterion, it is likely that some proportion of respondents have a pharmacy degree (or the 4.4% with a PhD) who do not practice and would be less likely to have obtained the information being asked about in the survey.
Response
Thank you for the insightful feedback.
We acknowledge that asking about students' employment status could have provided additional context, and we will consider including this in future studies.
As for limiting the survey to practicing pharmacists, we intended to include both pharmacists and pharmacy students to assess their knowledge and preparedness for managing monkeypox. While practicing pharmacists bear direct responsibility for patient care, pharmacy students represent the future workforce and will soon be involved in managing such cases. Including them allowed us to capture the broader spectrum of knowledge and readiness, providing insights into gaps in both current and future pharmacists' understanding of monkeypox. However, we agree that differentiating between active practitioners and non-practicing individuals (such as those with a PhD) could have further refined the analysis, and we will consider this in future research designs.
* The following paragraph was added to the manuscript "Future Directions"
"Future research should explore the differentiation between active practitioners and non-practicing individuals, such as those holding a PhD. This stratification could en-hance the analysis and provide more targeted insights regarding the knowledge and practices related to monkeypox management among pharmacists."
Comment
17) Lines 141-144: What is a “workshop” in this context? Are pharmacists required to attend these workshops? Did this specifically focus on monkeypox? How does this question relate to the study topic?
Response
In this context, the term "workshop" refers to continuing professional development (CPD) sessions that pharmacists may attend to enhance their knowledge on several health-related topics, including emerging diseases. These workshops are not mandatory but are often encouraged to maintain up-to-date knowledge in the field.
Regarding the focus on monkeypox, while the survey did not specify whether the workshops attended by participants were directly related to monkeypox, the question aimed to explore the general educational engagement of pharmacists. By understanding their participation in CPD activities, we hoped to assess whether a higher frequency of attending such sessions might correlate with better preparedness or knowledge related to emerging infectious diseases, such as monkeypox.
- The point was clarified in the methods section as follows:
"The initial section consisted of 7 items aimed at assessing participants' demographic characteristics. One of the questions in this first section asked participants about the number of workshops they attended, referring to continuing professional development, to gauge the overall educational engagement of pharmacists"
2. It was emphasized in the manuscript (result section) that these workshops were on general topics.
"Regarding attending workshops that covered various general topics, above one-quarter of the study participants (32.6%, n=126) reported not attending any general workshop per year, 16.6% attended one workshop (n=65), 21.5% attended two work-shops (n=84), 19.3% attended three workshops (n=75), 5.7% attended four workshops (n=22), and 4.2% attended 5 workshops or more (n=16)."
Comment
18) Line 171: Why is a mean reported when a median was used to stratify TK scores into a binary (high/low) variable?
Response
Thank you for your comment.
The data presentation has been revised.
The following sentence: "These scores had a mean of 6.48 (SD=4.55) on a scale…"
was changed as follows: "These scores had a median of 6.00 (IQR=6.00) on a scale… "
Comment
19) Lines 173-174 and 177-178: Why report frequencies of total scores rather than just a median, which is a more effective and reader-friendly way to present the same data?
Response
- The following two sentences were deleted:
- "In addition, the highest percentage was reported on scores 6 (n=37) and 7 (n=37), accounting for 6.3 for each"
- "Moreover, the highest percentage was reported on scores 1 (n=91) accounting for 23.45%"
- The following sentence: "The range of total knowledge scores (TK-score) among participants varied from -6 to 23. These scores had a mean of 6.48 (SD=4.55) on a scale from -26 (lowest possible score) to 26 (highest possible score). Surprisingly, none of the participants answered all knowledge items correctly. In addition, the highest percentage was reported on scores 6 (n=37) and 7 (n=37), accounting for 6.3 for each (Figure 3).
The participants' symptom knowledge scores (SK-score), measured on a scale from -9 to 9, ranged from -3 to 9, with an average of 2.45 (SD=2.39). It is worth men-tioning that only eight participants answered all the symptom knowledge correctly (2.06%). Moreover, the highest percentage was reported on scores 1 (n=91) accounting for 23.45% (Figure 3).
was changed as follows: "The range of total knowledge scores (TK-score) among participants varied from -6 to 23. These scores had a median of 6.00 (IQR=6.00) on a scale from -26 (lowest possible score) to 26 (highest possible score). Surprisingly, none of the participants answered all knowledge items correctly (Figure 3).
The participants' symptom knowledge scores (SK-score), measured on a scale from -9 to 9, ranged from -3 to 9, with a median of 2.00 (IQR=3.00). It is worth mentioning that only eight participants (2.06%) answered all the symptom knowledge correctly (Figure 3)."
Discussion
Comment
20) Line 226: Is “disproved” the right term to use? That typically refers to scientific evidence rather than survey responses.
Response
The following sentence: "The claim that monkeypox virus is a bacterial zoonotic infection, rather than viral, was disproved by the majority"
Was changed as follows:
"Most of the participants correctly identified that monkeypox is not a bacterial zoonotic infection"
Comment
21) Lines 228-233: This was not identified in the results as a key finding, yet it is presented here as being highly impactful. This discrepancy needs to be rectified.
Response
The following was deleted from the discussion section
"Regarding this, studies have shown that there is vertical transmission to the fetus during pregnancy (16–18). However, the proportion and the exact risk are yet unclear. Evidence suggests that the transmission is more likely during the third trimester than the first, especially if the mother has active maternal lesions, and there is also a higher risk of fatality during this period (16–18). "
Comment
22) Lines 240-241: Need a citation to support the statement that empathy enhances knowledge retention and that females exhibit more empathy than their male colleagues particularly within the health care setting.
Response
Done. References were added to the point
Comment
23) Lines 253-254: Impact it in what way? Interpret the results rather than just restating them.
Response
The current paragraph "The current study also assessed the pharmacists' perceptions of the monkeypox virus, 38.7% of them expressed concern about monkeypox becoming a future epidemic, and about a quarter believed that people vaccinated against COVID-19 are at higher risk of facing severe symptoms from monkeypox. These perceptions may impact the proactive engagement of pharmacists in monkeypox prevention and management."
Was edited as follows:
"The current study assessed the pharmacists' perceptions of the monkeypox virus. Notably, 38.7% expressed concern about monkeypox becoming a future epidemic, which could lead to increased vigilance in monitoring cases and educating patients about prevention measures. Additionally, about a quarter of the participants believed that people vaccinated against COVID-19 are at higher risk of experiencing severe symptoms from monkeypox. This perception could drive pharmacists to enhance their educational outreach and patient counseling."
Comment
24) Lines 258-260: How do you think these limitations could have potentially impacted study findings? Was anything done to try and mitigate these limitations?
Response
The paragraph was edited as follows:
"One of the study's limitations lies in its sampling methods, the likelihood that certain individuals are more likely to complete the study's survey than others. Potential participants decide independently whether to take part, leading to "self-selection bias". Self-selection bias may limit the generalizability of the study's findings, potentially af-fecting the accuracy of conclusions about pharmacists' perceptions and knowledge of monkeypox. To address this, the survey was distributed through various professional networks to capture a broader range of perspectives"
Reviewer 4 Report
Comments and Suggestions for Authors
The work is very interesting and deals with a very important topic. The results were presented in a nice, simple and understandable way.
I have a few suggestions to improve the manuscript.
- In the methods, it should be described in one sentence how many questions are included in each part of the questionnaire (the questionnaire has 3 parts).
- In a paragraph of 141 lines, it should be emphasized what the workshops refer to (workshops related to monkeypox or in general).
- In the text, the reference to Figure 1 is missing before the figure.
- Figure 2 is inserted in the table. This will be much clearer and easier to understand. One column can contain all questions, the second N (%) and the third the correct answers.
- In Figure 3, add a range from -26 to 26 in the first part, as in the second part of the figure (range from -9 to 9).
- In Table 3, remove the dots (• Male • Female; Single/ Divorced/ Widowed • Married…).
Author Response
Comment
The work is very interesting and deals with a very important topic. The results were presented in a nice, simple and understandable way.
I have a few suggestions to improve the manuscript.
Response
Thank you. Your time and effort in reviewing the manuscript are highly appreciated.
Comment
- In the methods, it should be described in one sentence how many questions are included in each part of the questionnaire (the questionnaire has 3 parts).
Response
Done. The number of items in each section was added.
"The initial section consisted of 7 items aimed at assessing participants' demographic characteristics. One of the questions in this first section asked participants about the number of workshops they attended, referring to continuing professional development, to gauge the overall educational engagement of pharmacists. The second section covered eight sources from which participants obtained information about monkeypox, alongside 26 items evaluating their knowledge items. The last section consisted of five items designed to evaluate participants' self-reported ability to diagnose and manage monkeypox possible cases, along with their perceptions regarding the future implications of the disease."
Comment
- In a paragraph of 141 lines, it should be emphasized what the workshops refer to (workshops related to monkeypox or in general).
Response
Done. It was emphasized that the workshops were general.
"Regarding attending workshops that covered various general topics, above one-quarter of the study participants (32.6%, n=126) reported not attending any general workshop per year, 16.6% attended one workshop (n= 65), 21.5% attended two work-shops (n= 84), 19.3% attended three workshops (n= 75), 5.7% attended four workshops (n= 22), and 4.2% attended 5 workshops or more (n= 16)."
Comment
- In the text, the reference to Figure 1 is missing before the figure.
Response
Thank you for bringing our attention to this critical point.
The reference in the text was added.
"The survey revealed that participants primarily relied on the WHO reports for monkeypox information (Figure 1), with a utilization rate of 79.8%. Followed closely are published research (74.7%), and government awareness campaigns (68.5%)."
Comment
- Figure 2 is inserted in the table. This will be much clearer and easier to understand. One column can contain all questions, the second N (%) and the third the correct answers.
Response
Thank you for your suggestion. While I understand that presenting the information in a table might enhance clarity, I believe that keeping it as a figure allows for a more visual and engaging representation of the data, which aids in quickly identifying patterns.
Comment
- In Figure 3, add a range from -26 to 26 in the first part, as in the second part of the figure (range from -9 to 9).
Response
Done. The range was added.
Comment
- In Table 3, remove the dots (• Male • Female; Single/ Divorced/ Widowed • Married…).
Response
Done
Round 2
Reviewer 2 Report
Comments and Suggestions for Authors
Thank you for replying to my comments. Everything is good now.
Author Response
Thank you for your positive feedback and important insights along the process.